# Cell Imaging Using Two-Photon Excited CdS Fluorescent Quantum Dots Working within the Biological Window

**DOI:** 10.3390/nano9030369

**Published:** 2019-03-05

**Authors:** Nannan Zhang, Xiao Liu, Zhongchao Wei, Haiying Liu, Jie Peng, Liya Zhou, Hongmei Li, Haihua Fan

**Affiliations:** 1Guangdong Provincial Key Laboratory of Nanophotonic Functional Materials and Devices, School of Information and Optoelectronic Science and Engineering, South China Normal University, Guangzhou 510006, China; clzhangnannan@163.com (N.Z.); shawer1004@163.com (X.L.); wzc@scnu.edu.cn (Z.W.); hyliu@scnu.edu.cn (H.L.); 2Department of Electronic Engineering, College of Information Science and Technology/College of Cyber Security, Jinan University, Guangzhou 510631, China; 3School of Chemistry and Chemical Engineering, Guangxi University, Nanning 530004, China; zhouliyatf@163.com; 4School of Life Sciences, Sun Yat-Sen University, State Key Lab for Biocontrol, Guangzhou 510275, China; lihongmei@mail.sysu.edu.cn

**Keywords:** CdS quantum dots, deep defect states, two-photon absorption, HePG2 cells, biological imaging

## Abstract

In recent years, two-photon excited semiconductor quantum dots (QDs) have been the subject of intense investigation due to their long excitation wavelength which helps to achieve deeper penetration and higher image resolution in optical bioimaging. In this paper, water-soluble CdS QDs were synthesized using a hydrothermal method and applied to human liver hepatocellular carcinoma (HepG2) cells. The first-principles calculation suggested that the S-rich defected structure contributes to a narrower band gap compared to the pristine structure. The resulting fluorescence wavelength was significantly red shifted, which was attributed to the deep defect states emission. The large Stokes shifts (> 200 nm) of the QDs can eliminate the possible cross-talk between the excitation light and the emission light. Two-photon induced red fluorescence emission can avoid overlapping with the autofluorescence emission of biological samples. The uptake and cell viability measurements of the HepG2 cells showed a good biocompatibility and a low toxicity of CdS QDs. Two-photon excited scanning microscopy images revealed that the HepG2 cells incubated with CdS QDs emitted bright red upconversion fluorescence and the fluorescence brightness was 38.2 times of that of the control group. These results support CdS QDs as a good candidate for application in cellular imaging.

## 1. Introduction

Two-photon absorption, a molecular excitation process by the simultaneous absorption of two photons, has attracted growing interest in many areas for its broad application in photonics and biophotonics such as microscopy, data storage, optical limiting, microfabrication [1,2,3,4], etc. Two photon absorption technology has many advantages, especially for bioimaging, such as minimum photo-damage to living organisms, high spatial resolution and contrast, and high light-penetration depth in tissues [5,6]. Quantum dots (QDs) are typically two-photon excited fluorescence nanomaterials. They have received much attention in the field of multiphoton bioimaging due to their superior photochemical stability and biological properties compared to other materials. First, it has been reported that the photochemical stability of QDs is 100 times higher than Rhodamine 6G (R6G) [7]. Therefore it can significantly overcome the photoquenching of organic imaging agents [8]. Second, the fluorescent QDs have a large surface-to-volume ratio, which is favorable for modifying the surface with biomolecules to achieve better biocompatibility, low toxicity, and a higher quantum yield (QY) [9].

As can be seen from previous references [10,11], the QD surface often creates surface or interfacial trap states that reduce the number of exciton states through (non)-radiative charge recombination. Brus and McLendon characterized the trap state fluorescence and concluded that the fluorescence is attributed to charge recombination between a trapped electron and a trapped hole at the QD’s surface [12,13]. Due to the appearance of defect states near the edges of the valence and conduction bands, the energy gap of the QDs becomes narrower than that of the intrinsic states [14]. Therefore, defective fluorescence shows a significant red shift compared to exciton emission by theoretical prediction. Veamatahau and Wei reported that controlling the size of QDs or surface-enriched elements can control the surface states of QDs and thus control the emission wavelength [11,15].

Among different types of QDs, CdS QDs have most commonly been studied owing to their ease of synthesis and distinctive optical properties. Since Chan [7] and Bruchez [16] demonstrated the usage of semiconductor QDs as fluorescent probes in cell, tissue, and organism imaging, multiphoton excited fluorescent QDs have been increasingly receiving researchers’ attention in applications of biological imaging [17], photodynamic therapy [18], and drug release control [13,14]. These studies motivated us to investigate high brightness, low-toxicity CdS QDs suitable for cell imaging.

In this paper, a type of surface S-rich CdS QDs was synthesized using a green hydrothermal method and applied in biological imaging. On the one hand, reducing the degradation of Cd ions from the surface of CdS QDs can diminish their biological toxicity [19]. Energy-dispersive X-ray spectrometry (EDS) and X-ray Photoelectron Spectroscopy (XPS) showed that there were more S atoms on the surface of the CdS QDs. The high uptake and high cell viability of the HepG2 cells cultured with the QDs showed a good biocompatibility and a low toxicity. On the other hand, we calculated the density of the states of intrinsic and defected CdS QDs using the density functional theory (DFT) method. The results show that the formation of the defect state is due to the introduction of excess sulfur on the surface, and the energy gap of the defect QD is narrower than the energy gap of the intrinsic QD. The CdS QDs exhibited a bright red defect state fluorescence centered at 640 nm, which produce a red-shift compared with the corresponding bulk band gap transition emission (510 nm) [20]. This result indicates that the fluorescence of the CdS QDs may be attributed to the recombination fluorescence of electrons and holes in defect states. The long wavelength of emission can effectively avoid the strong absorption of the biological sample and avoid contrast reduction due to enhanced emission in the autofluorescence wavelength range [5]. Two-photon absorption fluorescence was observed when the QDs were excited by an 800 nm femtosecond laser. The two-photon excited scanning microscopy images revealed that the upconversion fluorescence intensity of the HepG2 cells incubated with CdS QDs was 38.2 times of that of the control group. These results indicate that CdS QDs is a good bioimaging material and opens up a new path for the development of more bioimaging nanomaterials.

## 2. Materials and Methods

### 2.1. Synthesis of CdS QDs

The preparation of CdS quantum dots (QDs) was based on the previously reported synthesis method [21,22]. See the Appendix A for the detailed experimental process.

### 2.2. Transmission Electron Microscopy

The morphology and structure of the CdS QDs were characterized by transmission electron microscopy (TEM) images. TEM images, selected area electron diffraction (SAED) patterns, and EDS data of the CdS QDs were obtained, respectively, using TEM (JEM-2100HR, JEOL, Japan) equipped with energy-dispersive X-ray spectrometry (EDS) operating an accelerating voltage of 200 kV.

### 2.3. Optical Characterization

The absorption spectrum of the CdS QDs was measured using an ultraviolet–visible spectrophotometer (UV-2700, Shimadzu, Japan). The fluorescence spectrum of the CdS QDs was obtained using a fluorescence spectrophotometer (FL-2500, Hitachi, Tokyo, Japan). The XPS was measured at room temperature using an Al K-alpha X-ray Photoelectron Spectrometer (Escalab 250Xi, Thermo Fisher Scientific, Waltham, MA, USA). The fluorescence quantum yields (PL QYs) and the two-photon absorption cross-section (σ_2PA_) of the CdS QDs solution were measured. See the Appendix A for the detailed measurement. The two-photon fluorescence spectrum of the CdS QDs was obtained using a mode-locked Ti:sapphire oscillator (Mira 900S, Coherent, US), inverted microscope (Axio Observer A1, Zeiss, Berlin, Germany) and spectrometer (SR-500i-B1, Andor, Oxford, UK). The schematic diagram of the experimental setup is shown Appendix A. Femtosecond transient two-photon fluorescence decay was measured using a Fluorescence Lifetime Spectrometer (FL920, Kirkton, Dundee, UK). See the Appendix A for fluorescence lifetime measurement details.

### 2.4. Cell Viability and Cellular Uptake of CdS QDs

The human liver hepatocellular carcinoma (HepG2) cells used in our laboratory were obtained from the Cell Lab of the Cell Resource Center of the Chinese Academy of Sciences. The methods of cell viability and the uptake of CdS QDs of individual HepG2 cells were similar to those described in reference [23]. See the Appendix A for the detailed measurement methods.

### 2.5. Laser Scanning Microscopy

HepG2 cells were seeded into the culture dishes at 10^5^ cells/dish, and incubated for 24 hours in a humidified incubator (37 °C, 5% CO2). The cells were washed with phosphate buffer saline (PBS) and incubated with a fresh cell medium containing 0.16 mM CdS QDs. The cells were incubated for 6 hours in a humidified incubator (37 °C, 5% CO2) before being washed three times with PBS. A drop of fresh cell medium was added. Under the excitation of an 800 nm femtosecond pulsed laser, we obtained two-photon excited fluorescent images of the HepG2 cells using laser scanning microscopy (Nikon A1, Japan).

### 2.6. Numerical Simulations

First principle calculations based on the density functional theory (DFT) framework with the Nanodcal package were implemented in order to obtain the density of states (DOS) of the CdS QDs [24]. The Perdew–Burke–Ernzerhof (PBE) and general gradient approximation (GGA) functionals were used to optimize the geometric structure of the CdS QD [25] and the wavefunction was expanded by the double zeta plus polarization (DZP) basis which was set with an energy cutoff of 80 Hartree. The Brillouin zone was sampled with 4 × 4 × 4 Monkhorst and Pack k-points. A Cd_14_S_14_ cluster was constructed with a diameter of about 1 nm as a control, and then the corresponding defect states models were established by modifying the pristine Cd_14_S_14_ cluster, which included Cd_13_S_14_, Cd_14_S_15_, Cd_13_S_15_, Cd_14_S_13_, Cd_15_S_14,_ and Cd_15_S_13_ clusters.

## 3. Results and Discussion

### 3.1. First Principle Calculations Based on Density Functional Theory

The small-size QD surface often creates surface or interfacial trap states due to their higher surface-to-volume ratio, which would reduce the number of exciton states through (non)-radiative charge recombination. Therefore, surface trap emission is commonly observed in small CdS QDs [26,27,28]. We calculated the DOS of pristine and all possible defected structures using the Nanodcal package. The modeled structures and the corresponding DOS of CdS QDs are presented in Figure 1, which shows that the defect states narrow the intrinsic band gap by different levels. As shown in Figure 1c,e, Cd vacancies and S interstitials introduced some deep donor levels appearing near the valence band, which provided deep defect states recombination sites for CdS QDs. While the S antisites defect states in Figure 1g introduced some deep acceptor levels appearing near the conduction band. Cd-rich CdS QDs of S vacancies, Cd interstitials, and Cd antisites defect states also produced the same effect by introducing some deep acceptor levels near the conduction band, as shown in Figure 1d,f,h. The intrinsic band gap shown in Figure 1a is narrower than the experimental values, which is due to the underestimation of the band gap originating from the first-principles calculation algorithm [29].

### 3.2. Morphological and Structural Characterization

The synthesized CdS QDs aqueous solution was stored at room temperature for 24 months without precipitation or clustering, which shows good stability. Figure 2a shows the TEM image of the CdS QDs. The morphology of the CdS QDs is nearly a spherical shape and the particles are uniformly dispersed, and the average diameter is 3.4 nm, which is consistent with the diameter (*D*) calculation from formula (1) [30],
(1)$$D=-\left(6.6521\times 10^{-8}\right)\lambda^{3}+\left(1.9557\times 10^{-4}\right)\lambda^{2}-\left(9.2352\times 10^{-2}\right)\lambda +\left(13.29\right)$$
where λ refers to the first exciton absorption peak in the absorption spectrum of the CdS QDs. The lattice distribution of the CdS QDs is clearly shown, which indicates that CdS QDs have a good crystallinity. Figure 2b is the SAED pattern of CdS QDs. The polycrystalline diffraction rings correspond to the crystal planes of (111), (220), (200), and (311), respectively, which matches with that of the CdS cubic zinc blende structure [31]. Figure 1c depicts a high-resolution transmission electron micrograph (HRTEM) image of a CdS QD. The lattice fringe spacing is 2.89 Å, which corresponds to the cubic zinc blende structure (200) crystal plane of CdS. Figure 1d shows the EDS spectrum of the CdS QDs. The insert list of components of Figure 2d shows that the S atoms are slightly more than the Cd atoms. The stoichiometric mismatch of S and Cd elements shown in the EDS spectrum provides a possibility for the generation of a deep defect level [11,15].

The surface chemical composition significantly affects the fluorescence emission of the band edge [11]. Therefore, the surface composition of the CdS QDs was further determined by XPS [32,33] as shown in Figure 3. The XPS spectrum of the Cd 3d and S 2p regions obtained were fitted by Gaussian functions with the distinct core electron binding energies of cadmium and sulfur: cadmium 3d 5/2 and 3/2 electrons have their binding energies at 405 and 412 eV, respectively; sulfur 2p 3/2 and 1/2 electrons have their binding energies at 162 and 163 eV [34]. The experimental binding energy of sulfur 2p 1/2 was slightly larger than 163 eV. We attributed this disparity to S atoms from stabilizer molecules thioglycolic acid (TGA) forming S–S bonds with surface S atoms [35]. The ratio of Cd to S atoms was determined from the integrated Cd and S signals with the appropriate atomic sensitivity factors from the spectrometer. Elemental analysis data are summarized in Table 1, which shows that the number of S atoms was significantly higher than that of the Cd atoms on the surface.

### 3.3. Optical Properties

Figure 4a shows the ultraviolet–visible absorption spectrum and fluorescence spectrum of the CdS QDs with a 360 nm excitation wavelength. The first exciton absorption peak is 400 nm, shorter than the absorption peak of the CdS bulk material (510 nm) [22]. The significant blue shift is caused by the quantum confinement effect of QDs [36]. However, the peak of the fluorescence is 640 nm. The exciton emission peak at 460 nm is extremely weak. We attributed the fluorescence peak at 640 nm to the recombination of a trapped electron and a trapped hole in the defect levels [15,37].

An 800 nm femtosecond pulsed laser was applied to the CdS QDs to study the upconversion fluorescence (UPF). The UPF peak is at 640 nm, which is consistent with the corresponding fluorescence emission, as shown in Figure 4b. Generally, the intensity of UPF depends on the power of the excitation laser. To further understand the mechanism of UPF in the CdS QDs, we obtained UPF spectrums with different laser powers and plotted them in logarithms. The logarithm of the intensity and the laser power exhibited a linear relationship. According to the formula I~P^K^ [38], where I is the fluorescence intensity of the UPF spectrum, and P is the excitation power of the pulsed laser, and k is the exponent of the power function, the fitted k is 2.06, which indicates that the excitation generates two-photon fluorescence. The schematic diagram of the two-photon excitation process is shown in Figure 4d. The fluorescence quantum yield is about 8.14%, which is higher than that of CdS QDs prepared by non-hot-injection one-pot approaches [39]. The two-photon absorption cross-section and fluorescence lifetime of the CdS QDs solution are shown in Table 2. 

### 3.4. Cellular Toxicity of CdS QDs in HepG2 Cells

Biocompatibility must be considered when the CdS QDs are applied to cellular imaging. We incubated HepG2 cells with different concentrations for 12 and 24 hours, and measured the viability by 3-(4, 5-dimethylthiazol-2-yl)-2, 5-diphenyltetrazolium bromide (MTT) assay. The results are shown in Figure 5. It can be seen from Figure 5 that the CdS QDs can be considered to be nontoxic to HepG2 cells (concentration below 160 μM), as the viability of the HepG2 cells still exceeds 85% at a concentration of 160 μM after being incubated with the CdS QDs for 24 hours. The concentration and the viability were higher than CdSe/ZnS QDs, which are 5 nM and 83%, respectively [40]. Appendix A shows that the morphology of the cells changes very little within 24 hours when the concentration does not exceed 160 μM. When the concentration is lower than 160 μM, the cell viability is maintained above 85%. However, at 200 μM, the 24 hours cell viability was reduced to less than 70%, so we used concentrations below 160 μM in the following experiments.

### 3.5. Interaction between the CdS QDs and HepG2 Cells

To ascertain the cellular uptake and the intracellular distribution of CdS QDs in HepG2 cells, TEM images of cell slices were applied after cell incubation for 24 hours. As shown in Figure 6a and 6b, the CdS QDs entered into the interior of the organelles and agglomerated into clusters of the cell vesicles through encapsulation. We obtained the whole cell TEM diagram for observing the distribution of QDs in the cell, as shown in Appendix A. Figure 6c is the EDS spectrum of CdS QDs clusters inside the cell. It can be seen from the map that these nanoparticles in the cells were CdS QDs. The atomic Cd/S ratio resulting from the EDS map is 0.5, which is due to the organics that make up the cells containing S element. Os, Pb, and other elements apart from Cd and S are found in the EDS map due to the additional chemicals and nickel meshes used in the process of preparing the cell slices.

The cellular internalization process plays a crucial role in bioimaging of nanoparticles. Using inductively coupled plasma-mass spectrometry (ICP-MS) (ICAP-qc, Thermo Fisher, Germany), we further measured the cellular uptake in a solution with a concentration of 160 uM. The molar mass and the uptake for a single HepG2 cell that we obtained for the 3.4 nm CdS QDs were 5.97 × 10^4^ g/mol and 3.03 × 10^−12^ g respectively, resulting in 3.06 × 10^7^ particles of CdS QDs intake for each cell.

### 3.6. Two-Photon Excitation Bioimaging

We studied the application of the water-soluble CdS QDs in two-photon fluorescence bioimaging. After incubating HepG2 cells with 160 uM CdS QDs in a culture dish for 6 hours, the HepG2 cells were washed three times with PBS. Then, the HepG2 cells were scanned by a laser scan microscope excited with an 800 nm femtosecond pulsed laser and the two-photon fluorescence images were captured. Figure 7a–d shows the bright field image, the images of green and red channels and combination image of the control group, respectively. Figure 7e–h shows the bright field image, the images of the green and red channels and the superposition image of the experimental group, in which the experimental group and the control group were incubated with and without CdS QDs, respectively. When the cells were excited by an 800 nm laser, the upconversion fluorescence intensity of the cells was recorded by the photomultiplier tube of each channel in the microscopy. When the laser scanned the cells, the two-photon excited imaging was obtained. The experimental group emitted bright red fluorescence, indicating that the CdS QDs had entered into the cells, while the control group HepG2 cells emitted only green auto-fluorescence [41]. As can be seen from Figure 7g, there are many bright spots in the cells. Combined with the TEM image, it can be inferred that these bright spots are luminescence of intracellular quantum dots. In addition to the bright spot, red fluorescence can also be observed in other parts of the cell due to the strong scattering effect of various organelles and cell fluids within the cell. Red light emitted by quantum dots is scattered by other organelles or cell fluids, so red light can be observed in other parts. The fluorescence brightness of the selected cells is listed in Table 3. The fluorescence brightness of the experimental group was 38.2 times of that of the control group that was incubated without CdS QDs. Besides, the similar fluorescence brightness enhancement can be achieved in other cells. See the Appendix A for detail measurements. These results further support the superior properties of two-photon fluorescence cellular imaging using water-soluble CdS QDs.

## 4. Conclusions

In this paper, a type of surface S-rich CdS QDs was designed and synthesized. The first-principles simulation results indicate that the defected QDs have a smaller band gap, which corresponds to a longer emission wavelength compared with the band gap of the pristine QDs. The characterization results proved the stoichiometric mismatch of the S and Cd elements in the QDs that we synthesized. The emission wavelength of the CdS QDs was longer than that of the intrinsic emission in CdS bulk, which indicates a narrower band gap caused by the defected structure. Red two-photon fluorescence was generated by a two-photon absorption process when an 800 nm laser excited the CdS QDs. The CdS QDs were further investigated for their fluorescence brightness, biocompatibility and toxicity. When the HepG2 cells were incubated with CdS QDs, the two-photon excited fluorescence brightness of the cells was 38.2 times of that of the control group. The uptake and cell viability were 3.06 × 10^7^ CdS QD particles for each cell and 85% incubated with 160 μM CdS QDs. The results show that the water-soluble CdS QDs from a hydrothermal synthesis are promising as a candidate for cellular bioimaging application.

## Figures and Tables

**Figure 1 nanomaterials-09-00369-f001:**
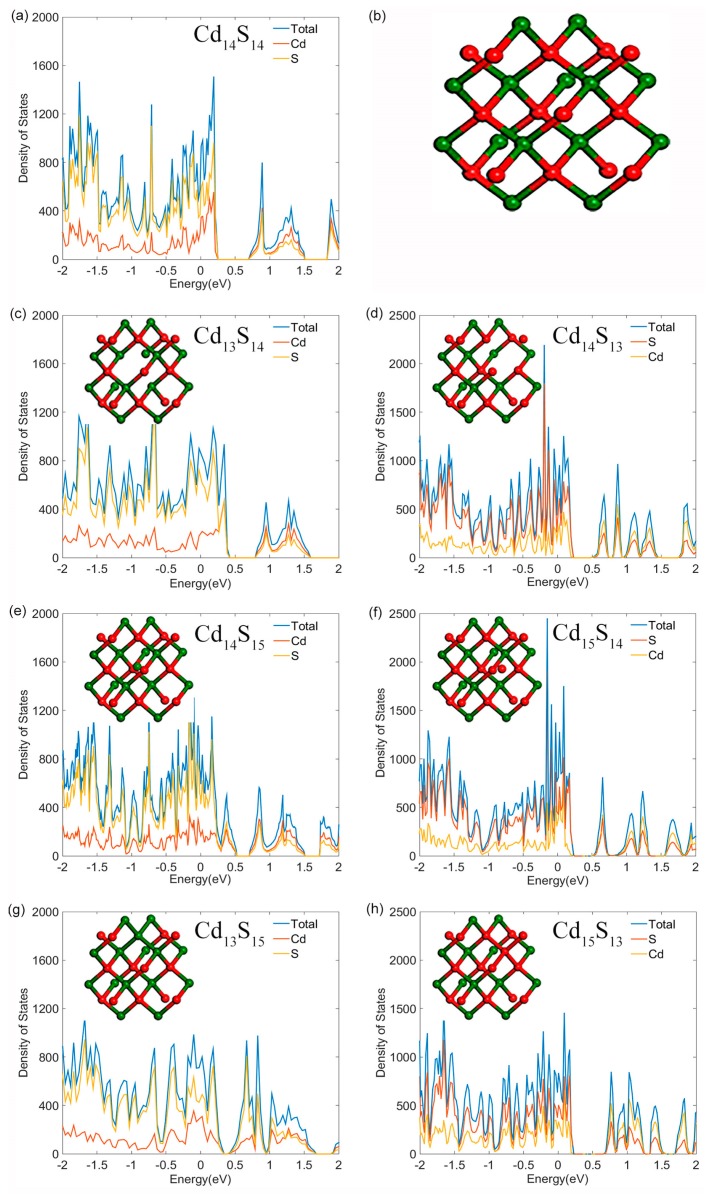
Density of states (DOS) of (**a**) pristine CdS quantum dots (QDs); (**b**) pristine CdS QD model; (**c**,**e**,**g**) represent DOS of CdS QDs with cadmium vacancies, sulfur interstitials and sulfur antisites, respectively; (**d**,**f**,**h**) represent the DOS of CdS QDs with sulfur vacancies, cadmium interstitials and cadmium antisites, respectively. The corresponding models are inserted in the DOS diagrams, respectively. S and Cd atoms are represented by green and red balls, respectively.

**Figure 2 nanomaterials-09-00369-f002:**
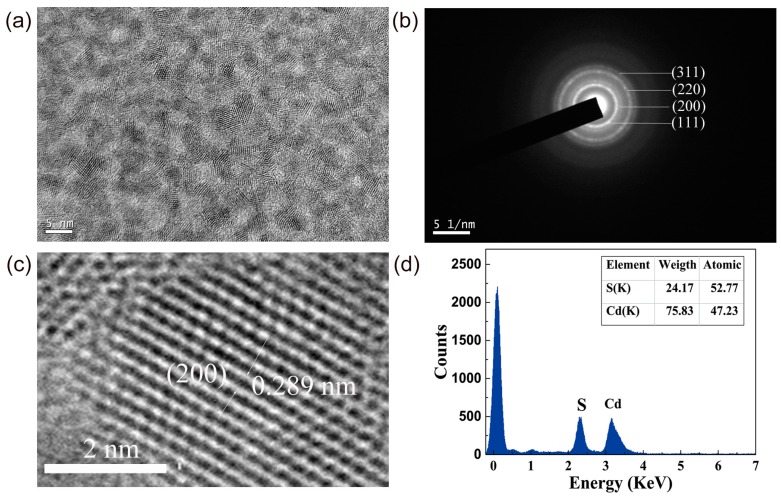
(**a**) TEM image of the synthesized CdS quantum dots (QDs); (**b**) SAED pattern of CdS QDs, the relevant crystal planes are indexed in the figure; (**c**) HRTEM image of a single QD that displays (200) crystal plane lattice fringes; (**d**) EDS spectrum of CdS QDs in our experiments.

**Figure 3 nanomaterials-09-00369-f003:**
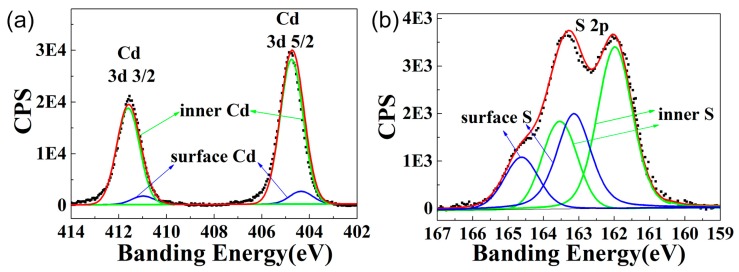
XPS elemental analysis of CdS QDs. (**a**) and (**b**) are XPS counts of Cd 3d electrons and S 2p electrons, respectively. Black dots represent raw XPS data, green lines are Gaussian curve fits at energies corresponding to core atoms, blue lines are Gaussian curve fits centered at energies corresponding to electrons from atoms on the surface, and red lines represent convolution of blue and green lines. Cd signals included contributions from Cd 3d 3/2 and 5/2 electrons, S signals included contributions from S 2p 1/2 and 3/2 electrons.

**Figure 4 nanomaterials-09-00369-f004:**
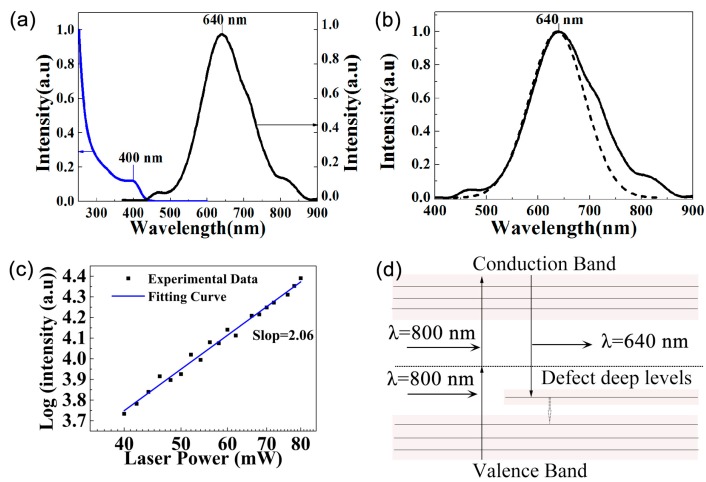
(**a**) Absorption and fluorescence spectrum of the CdS QDs, the blue line and black line represent the absorption spectrum and fluorescence spectrum, respectively; (**b**) Fluorescence and two-photon excited fluorescence spectrum of the CdS QDs, the solid line and dash dot line represent the fluorescence spectrum and two-photon excited fluorescence spectrum, respectively; (**c**) Power dependence of two-photon excited fluorescence intensity of CdS QDs on 800 nm excitation power; (**d**) The schematic diagram of the two-photon absorption process induced deep defect states fluorescence.

**Figure 5 nanomaterials-09-00369-f005:**
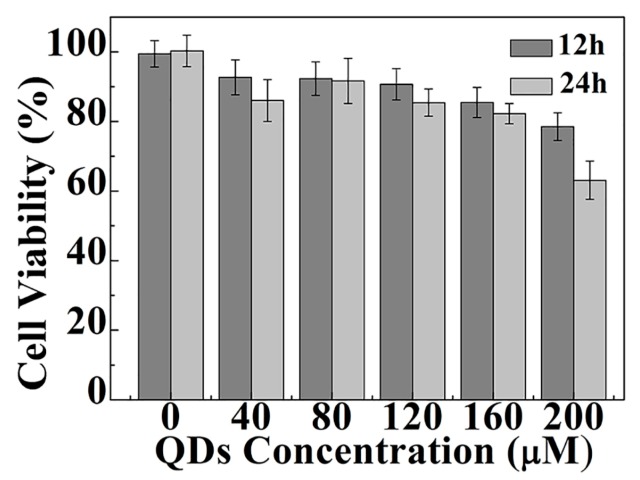
The viability of HepG2 cells with different concentrations of CdS QDs measured by MTT assay. Error bars represent the standard deviation of quadruple experiments.

**Figure 6 nanomaterials-09-00369-f006:**
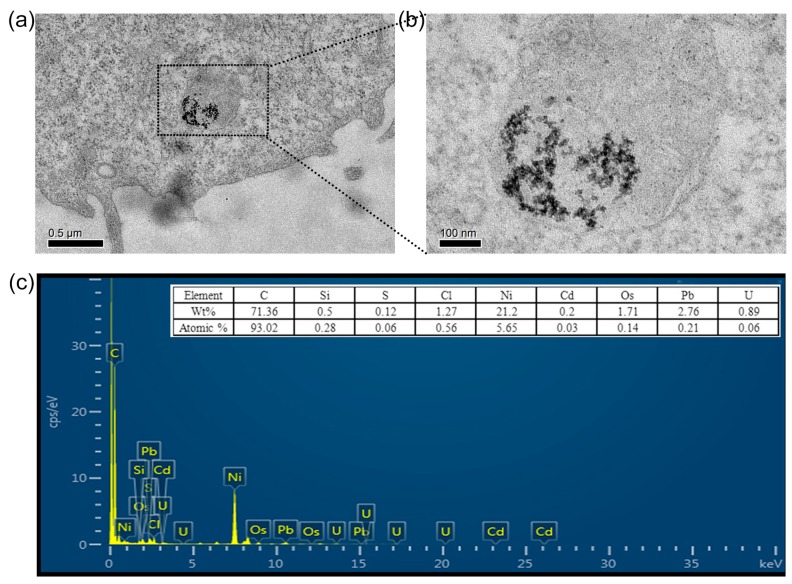
(**a**) TEM image of a HepG2 cell incubated with CdS QDs for 24 hours; (**b**) The enlarged TEM image of CdS QDs naturally created in the lysosome of the HepG2 cell; (**c**) The EDS spectrum of CdS QDs inside the HepG2 cell.

**Figure 7 nanomaterials-09-00369-f007:**
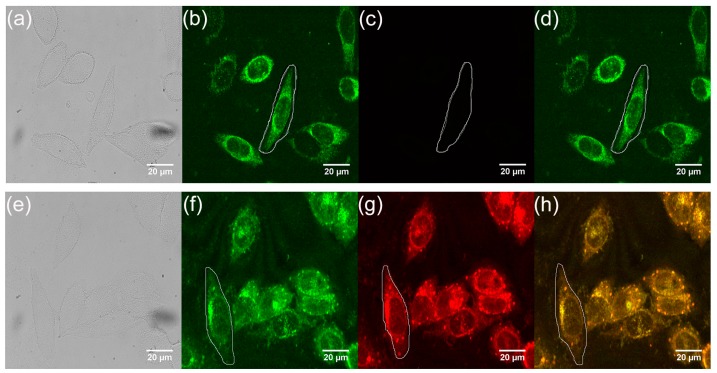
(**a**–**d**) The image of control group HepG2 cells without culturing with CdS QDs; (**a**) The bright field image; (**b**) The two-photon absorption fluorescence images with the green channel; (**c**) The red channel; (**d**) The combination image of green and red channels; (**e**–**h**) The images of HepG2 cells incubated with CdS QDs for 6 h; (**e**) The bright field image; (**f**) Two-photon absorption fluorescence images with the green channel; (**g**) The red channel; (**h**) Combination image of green and red channels.

**Table 1 nanomaterials-09-00369-t001:** XPS elemental analysis of CdS QDs. The table shows the atomic percentage of elements from raw XPS data. Surface-to-inner ratios of Cd and S (Cds/Cdi and S_s_/Si) were calculated from the integrated peak area fitted from Cd and S signals in the XPS data.

QDs	C1s%	Cd3d%	S2p%	Cd/S Ratio	Cd_s_/Cd_i_	S_s_/S_i_	Cd_s_/S_s_
CdS	85.61	6.50	7.89	0.83	0.086	0.5	0.2

**Table 2 nanomaterials-09-00369-t002:** Quantum efficiency, two-photon absorption cross-section and fluorescence lifetime of the CdS QDs measured by R6G.

Sample	$\lambda_{\mathrm{ex}}^{a}$ (nm)	$\lambda_{\mathrm{ex}}^{b}$ (nm)	$\lambda_{\mathrm{em}}$ (nm)	Φ (%)	UCL Γ (ns)	σ_2_ (GM)
**R6G**	425	800	553	95	-	39.90
**CdS**	425	800	640	8.14	1.68	54.33

^a^ The excitation wavelength of 425 nm was used to measure the quantum efficiency of the CdS QDs; ^b^ An excitation wavelength of 800 nm was selected to measure the two-photon absorption-cross of the CdS QDs.

**Table 3 nanomaterials-09-00369-t003:** The fluorescence brightness of the selected cell.

Sample	(b)	(f)	(c)	(g)	(d)	(h)
Area (cell)	1	1	1	1	1	1
Intensity (min)	11	9	1	9	11	96
Intensity (max)	110	99	3	89	110	179
Intensity (mean)	37.7	31.2	1.0	38.2	37.7	133.2

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
