# Peer review of "Cell Imaging Using Two-Photon Excited CdS Fluorescent Quantum Dots Working within the Biological Window"

_nanomaterials, 2019, doi:10.3390/nano9030369_

Round 1

Reviewer 1 Report

The authors describe the synthesis and characterization of CdS QDs. In addition, the cell viability, cellular viability, and two-photon absorption process were described. The synthesis of CdS QDs is not novel and the cytotoxicity of Cd-based QDs have been heavily scrutinized in the literature already. Overall the manuscript appears rushed and needs extensive editing.

Authors should edit the manuscript as there are errors throughout, for example, line 49 as misspelled references.

Acronyms should be used throughout the manuscript after they have been defined. They should be defined when they are first used. For example, XPS is written in full, “x-ray photoelectron spectroscopy” and “XPS”. This needs to be checked throughout.

line239-What is meant by “The concentration was almost free of toxic to HePG2 cells”

There should be spaces between values and their units

The caption for Figure 7 states “(e)-(h) The images of HepG2 cells incubated with GNRs for 6 h”-is GNR supposed to be CdS QDs?

The authors describe the cellular uptake of the CdS QDs using TEM. The CdS QDs appear to completely decorate the cells (inside and out) but this does not compare to the TEM images were they are located within endosomes. Please explain.

Author Response

We would like to thank the comments and suggestions of the reviewer which are definitely helpful for improving the quality of our manuscript. The responses (abbreviated as R) to these comments (abbreviated as C) and the changes made in the revised manuscript (marked in red colour) are described in detail in the following.

C1: The authors describe the synthesis and characterization of CdS QDs. In addition, the cell viability, cellular viability, and two-photon absorption process were described. The synthesis of CdS QDs is not novel and the cytotoxicity of Cd-based QDs have been heavily scrutinized in the literature already. Overall the manuscript appears rushed and needs extensive editing.

R1: We are very sorry for the unclear expressions in the manuscript that lead to misunderstandings in the process of review. Here we rewrite the corresponding parts of manuscript in a more clear way as following to make the originality prominent (also revised in the manuscript).

C2: Authors should edit the manuscript as there are errors throughout, for example, line 49 as misspelled references.

R2: According to the suggestion of the reviewer, English language and style have been carefully examined. The line 49 has been revised to: “references”.

C3: Acronyms should be used throughout the manuscript after they have been defined. They should be defined when they are first used. For example, XPS is written in full, “x-ray photoelectron spectroscopy” and “XPS”. This needs to be checked throughout.

R3: According to the suggestion of the reviewer, the manuscript has been checked throughout.The “XPS” have been written in full “x-ray photoelectron spectroscopy”(line 71).

C4: The line 239-What is meant by “The concentration was almost free of toxic to HePG2 cells”

R4: In the line 239 “the concentration was almost free of toxic to HePG2 cells” refer to 160 uM  CdS QDs exhibit weak toxicity for HePG2 cells. This sentence has been revised to: “The 160 uM  CdS QDs  exhibit weak toxicity for HePG2 cells”.

C5: There should be spaces between values and their units

R5: According to the suggestion of the reviewer, the line 178 has been revised to 163 eV. The line 83 has been revised to 800 nm and the figures have been revesed. 

C6: The caption for Figure 7 states “(e)-(h) The images of HepG2 cells incubated with GNRs for 6 h”-is GNR supposed to be CdS QDs?

R6: According to the suggestion of the reviewer, the caption for Figure 7 have been revised to “ (e)-(h) The images of HepG2 cells incubated with CdS QDs for 6 h”. 

C7: The authors describe the cellular uptake of the CdS QDs using TEM. The CdS QDs appear to completely decorate the cells (inside and out) but this does not compare to the TEM images were they are located within endosomes. Please explain.

R7: Figure 6a and b are the TEM images of cells cultured with quantum dots for 24 hours. Combining with information from TEM images and EDX results, it was confirmed that after 24 hours of co-culture with HepG2 cells, CdS QDs were taken up by cells and accumulated in vesicular organelles after entering into cells.  The TEM image of the whole cell was also obtained in this work as showed in FigureS3 below.  It can be seen that the quantum dot clusters are distributed in the organelles inside the cells.  Figure7g is the two-photon excited fluorescence image of the cells cultured with the CdS QDs. As can be seen from Figure 7g, there are many bright spots in the cells. Combined with the TEM image, it can be inferred that these bright spots are the luminescence of intracellular quantum dots. In addition to the bright spot, red fluorescence can also be observed in other parts of the cell due to the strong scattering effect of various organelles and cell fluids within the cell. Red light emitted by quantum dots is scattered by other organelles or cell fluids, so red light can be observed in other parts. This part of the analysis has been added to section 3.6.

Reviewer 2 Report

The authors synthesized water-soluble, CdS quantum dots through a hydrothermal method and tested their biocompatibility and possibility of use for bioimaging via two-photon excitation. Due to the preparation method, the S-rich defected structure allowed for a lower toxicity, narrower gap and larger Stokes shift. The work is correctly structured, informative, scinetifically sounding and exhaustive. Experiments are sufficiently detailed and presented. Comparison with the results from numerical simulation gives plausible explanations for the observed effects. Conclusions are well supported by the experimental observations. Apart from some unclear sentences, missing comments and few typos, reported below, the paper is well written and easy to follow. Therefore, with some minor corrections and additional comments, the paper can be accepted.

1.       In line 178, the acronym TGA should be defined;

2.       In line 203, the authors should indicate the excitation wavelength of the fluorescence spectrum of Fig. 4a;

3.       In line 238, the authors should comment  how the present results about toxicity compare with those of the previous works, without asking the reader to necessarily check in the literature;

4.       This should also be done in the Conclusion section;

5.       In line 251, the authors stated that “It can be seen from the map that these nanoparticles in the cells were CdS QDs.”. However, the atomic Cd/S ratio resulting from the EDS map is 0.5 instead of  » 1, as expected. The authors should comment  this point;

As regards the typos:

In line 201, “The first exciton absorption peak is 400 nm, smaller than that of…”    ->   “The first exciton absorption peak is at 400 nm, at shorter wavelength than…..”

In line 215, “the quantum yield is about 8,14…”   ->   “is about 8,14 %……”

In line 238, “remains…”   ->   “remained..”

Author Response

We would like to thank the comments and suggestions of the reviewer which are definitely helpful for improving the quality of our manuscript. The responses (abbreviated as R) to these comments (abbreviated as C) and the changes made in the revised manuscript (marked in blue color) are described in detail in the following.

C1: The authors synthesized water-soluble, CdS quantum dots through a hydrothermal method and tested their biocompatibility and possibility of use for bioimaging via two-photon excitation. Due to the preparation method, the S-rich defected structure allowed for a lower toxicity, narrower gap and larger Stokes shift. The work is correctly structured, informative, scinetifically sounding and exhaustive. Experiments are sufficiently detailed and presented. Comparison with the results from numerical simulation gives plausible explanations for the observed effects. Conclusions are well supported by the experimental observations. Apart from some unclear sentences, missing comments and few typos, reported below, the paper is well written and easy to follow. Therefore, with some minor corrections and additional comments, the paper can be accepted.

R1: Thanks the suggestions of the reviewer. Here we rewrite the corresponding parts of manuscript in a more clear way as following to make the originality prominent (also revised in the manuscript).

C2: In line 178, the acronym TGA should be defined;

R2: According to the suggestion of the reviewer, English language and style have been carefully examined. The line 178 has been revised to: “ Thioglycolic acid (TGA)”.

C3: In line 203, the authors should indicate the excitation wavelength of the fluorescence spectrum of Fig. 4a;

R3: According to the suggestion of the reviewer, the line 203 has been revised to: “ Figure 4a shows the ultraviolet-visible absorption spectrum and fluorescence spectrum of the CdS QDs with a 360nm excitation wavelength.”

C4:  In line 238, the authors should comment how the present results about toxicity compare with those of the previous works, without asking the reader to necessarily check in the literature;

R4:According to the suggestion of the reviewer, the line 238 has been revised to : “The concentration and the viability are higher than CdSe/ZnS QDs, which are 5 nM and 83%, respectively [42].When the concentration of CdS QDs smaller than 160μM we can use the CdS QDs for cellular imaging”.

C5:  This should also be done in the Conclusion section;

R5: According to the suggestion of the reviewer, the cell viability has been revised to: “ The uptake and cell viability of are 3.06 × 107 CdS QD particles for each cell and 85% incubated with 160μM CdS QDs” in the Conclusion section.

C6:  In line 251, the authors stated that “It can be seen from the map that these nanoparticles in the cells were CdS QDs.”. However, the atomic Cd/S ratio resulting from the EDS map is 0.5 instead of  » 1, as expected. The authors should comment this point;

R6: According to the suggestion of the reviewer, the line 251 has been revised to: “It can be seen from the map that these nanoparticles in the cells were CdS QDs. The atomic Cd/S ratio resulting from the EDS map is 0.5, which is due to the organics that make up the cells ontain S element”.

C7:As regards the typos:In line 201, The first exciton absorption peak is 400 nm, smaller than that of…”    ->   The first exciton absorption peak is at 400 nm, at shorter wavelength thanIn line 215, the quantum yield is about 8,14…”   ->   is about 8,14 %……”In line 238, remains…”   ->   remained..

R7: According to the suggestion of the reviewer, the line 201 has been revised to: “ the first exciton absorption peak is at 400 nm, shorter than the absorption peak of the CdS bulk material”. The line 215 has been revised to:” the quantum yield is about 8.14%. The line 238 has been revised to the cell viability remained 85% after 24 hours”.

Reviewer 3 Report

The manuscript by Zhang et al. tackles a topic of interest to the Nanomaterials journal audience, namely the synthesis and characterization of two-photon excited CdS fluorescent quantum dots for cell imaging. However, this field of research is plenty of examples of similar QDs probes. The state of the art is incompletely represented and the originality of the reported work is not sufficiently clearly expressed to consider the paper publishable in its present form.

Minor comments:

1.     In the abstract is not clear what do the authors means for ‘pristine structure’ in comparison to S-rich defected structure; moreover, it is not clear at all how these S-rich defected structures were synthesized and selected with respect to the pristine structures.

2.     Is the 38.2 times increase of fluorescence brightness significant in comparison to other QDs probes already known from literature and/or available on the market?

3.     The are many typos and unclear phrasal forms in the paper, please carefully proof-read spell check to eliminate errors.

Some examples: line 46 (change ‘.secondly’ to ‘. Secondly’); line 47 (change ‘large surface area’ to ‘large surface-to volume ratio’); line 49 (change ‘refrernces’ to ‘references’); line 76 (change ‘longer’ to ‘red-shifted’); lines 115-116 (rephrase ‘The fresh cell medium containing 0.16 mM CdS QDs replaced the old cell medium’, for instance  ‘Cells were washed with fresh cell medium and incubated with …’; line 210 (change ‘spectrums’ with ‘spectra’).

4.     Lines 60-61: ‘However, the application in bioimaging has beenprevented because of their biological toxicity’ is not true.

5.     Figure 5 lacks error bars and significance.

6.     Table 3 is meaningless without statistical significance.

Author Response

We would like to thank the comments and suggestions of the reviewer which are definitely helpful for improving the quality of our manuscript. The responses (abbreviated as R) to these comments (abbreviated as C) and the changes made in the revised manuscript (marked in red color) are described in detail in the following.

The manuscript by Zhang et al. tackles a topic of interest to the Nanomaterials journal audience, namely the synthesis and characterization of two-photon excited CdS fluorescent quantum dots for cell imaging. However, this field of research is plenty of examples of similar QDs probes. The state of the art is incompletely represented and the originality of the reported work is not sufficiently clearly expressed to consider the paper publishable in its present form.

C1: In the abstract is not clear what do the authors means for ‘pristine structure’ in comparison to S-rich defected structure; moreover, it is not clear at all how these S-rich defected structures were synthesized and selected with respect to the pristine structures. Moreover, it is not clear at all how these S-rich defected structures were synthesized and selected with respect to the pristine structures.

R1: Thanks the suggestions of the reviewer. "Pristine structure" is the structure without defect. Similar usage can be found in the abstract in "Origin of the Broadband Photoluminescence of Pristine and Cu+/Ag+-Doped Ultrasmall CdS and CdSe/CdS Quantum Dots" Stroyuk, O et al, JOURNAL OF PHYSICAL CHEMISTRY C 122 18 10267-10277.

The S-riched defected structures were synthesized by green hydrothermal method. The characterization proofed the S-riched construction. The synthesis S-rich defected CdS QDs procedures have been summarized as: “100 mL of 0.02mol/L CdCl2•2.5H2O solution was loaded into a 250 mL the three-necked flask. 0.35 mL of thioglycolic acid (TGA) was added to three-necked flask during magnetic stirring. The pH value of the mixed solution was adjusted to be 10.5 by the dropwise addition of 1 mol/L NaOH solution. Then 0.2459 g of NaS•9H2O was dissolved into 5 mL of deionized water and added to the above mixture. Finally, the solution mixture was heated to 100 ° C in an oil bath and refluxed for 5 hours before being cooled to room temperature. A transparent pale yellow CdS QDs solution was obtained.” The S-rich defected CdS QDs were formed by adding 0.35 mL of TGA. In order to reduce the repetition rate of the manuscript, the synthesis process is written into the supplementary material.

C3: Is the 38.2 times increase of fluorescence brightness significant in comparison to other QDs probes already known from literature and/or available on the market?

R3: The comparison of the fluorescence intensity as a digital form between the cells incubated with QDs and control group (incubated without QDs) hasn’t been reported in other literatures, as far as we know. Multi-photon excited CdS fluorescent quantum dots using as the probes for cell imaging gets a lot of attentions. Many papers studied the two-photon absorption cross-section of the QDs(Two-Photon Absorption in Penicillamine Capped CdS Tetrapods, J. Mater. Chem. C; D. Wawrzyńczyk , 2017, 5, 1724--1729 ; Wavelength dependence of nonlinear optical properties of colloidal CdS quantum dots, Janusz Szeremeta et al, Nanoscale, 2013,5, 2388-2393 ). Other Papers focus on the bio-compatibility and bio-sensor of the QDs ( Graphene Oxide Nanoparticles as a Nonbleaching Optical Probe for Two-Photon Luminescence Imaging and Cell Therapy , Jing-Liang Li et al, Angew. Chem. Int. Ed. 2012, 51, 1830 –1834; Two-Photon Fluorescent Molybdenum Disulfide Dots for Targeted Prostate Cancer Imaging in the Biological II Window, Carrie Sweet et al, ACS Omega 2017, 2, 1826−1835 ). The focus of our paper is to get bioimaging material with high brightness, low-toxicity and good biocompatibility with simple synthesis methods through combining theoretical analysis with experiment results. The S-riched defect introduced in this paper is a simple way to obtain high uptake and high cell viability. The defect can adjust the wavelength of the emission at the same time maintain the intensity. Red emission is an important fluorescence channel besides the green auto-fluorescence of the cell. Comparing with other literatures, the two-photon absorption cross section of quantum dots used in this paper is not the highlight. However, the combination of the former two aspects can also achieve a good fluorescence imaging effect, which can increase the brightness of the red channel of cells cultured with quantum dots by 38.2 times.

C4: There are many typos and unclear phrasal forms in the paper, please carefully proof-read spell check to eliminate errors.

Some examples: line 46 (change ‘.secondly’ to ‘. Secondly’); line 47 (change ‘large surface area’ to ‘large surface-to volume ratio’); line 49 (change ‘refrernces’ to ‘references’); line 76 (change ‘longer’ to ‘red-shifted’); lines 115-116 (rephrase ‘The fresh cell medium containing 0.16 mM CdS QDs replaced the old cell medium’, for instance  ‘Cells were washed with fresh cell medium and incubated with …’; line 210 (change ‘spectrums’ with ‘spectra’).

R4: According to the suggestion of the reviewer, the line 46 has been revised to “Secondly”; the line 47 has been revised to “large surface-to volume ratio”; the line 49 to “references”; the line 76 has been revised to “red-shifted”; the lines 115-116 have been revised to “Cells were washed with phosphate buffer saline (PBS) and incubated with the fresh cell medium containing 0.16 mM CdS QDs”; the line 210 has been revised to “spectra”.

C5: Lines 60-61: ‘However, the application in bioimaging has been prevented because of their biological toxicity’ is not true.

R5: According to the suggestion of the reviewer, the sentence in lines 60-61 :‘However, the application in bioimaging has been prevented because of their biological toxicity’ has been deleted

C6: Figure 5 lacks error bars and significance.

R6: According to the suggestion of the reviewer, error bars have been added in Figure 5.

C7: Table 3 is meaningless without statistical significance

R7: Thanks the reviewer for pointing out this. The purpose of Table 3 is to digitise figure 7. When the cells were excited by 800nm laser, the upconversion fluorescence intensity of the cells was recorded by the photomultiplier tube of each channel in the microscopy. When the laser scans the cells, the two-photon excited imaging can be obtained. The intensity in figure 7 represents the photocurrent in different channels. To illustrate the enhancement of the QDs in fluorescence, we compared with different cells in figure7 as showed in figureS5 and tableS1 below and added them in the supplementary materials.

Round 2

Reviewer 1 Report

The authors have addressed the comments from the reveiwers, however this is still editing that needs to be done. 

Line 49 - should be either “a quantum yield” or “quantum yields”

Equation (1) should be on one line (line 159)

Line 200 - shoudl be "360 nm"

What stastistical analysis was done for the cell viability?

The authors state "When the concentration of the CdS QDs was 160 uM, the cell viability remained 85% after 24 hours, which indicate that few cadmium ions dissociated from CdS QDs[21].". Beside being pooyl written and containing several mistakes, the authors cannot conclude the leaching of Cd ions into solution, subsequently reducing the cell viability. Further experiments would be needed to prove this. Several mistakes include u instead of the greek letter for micron, space bettween the reference and last word and poor English.

Author Response

We would like to thank the comments and suggestions of the reviewer which are definitely helpful for improving the quality of our manuscript. The responses (abbreviated as R) to these comments (abbreviated as C) and the changes made in the revised manuscript are described in detail in the following.

C1: Line 49 - should be either “a quantum yield” or “quantum yields”

R1: According to the suggestion of the reviewer, English language and style have been carefully examined. The sentence line 49 has been revised to :“ and a higher quantum yield (QY) [9]”.

C2: Equation (1) should be on one line (line 159)

R2: According to the suggestion of the reviewer, equation (1) has been on one line (line 159).

C3: Line 200 - should be "360 nm"

R3: According to the suggestion of the reviewer, the line 200 has been revised to: “360 nm”.

C4: What stastistical analysis was done for the cell viability? The authors state “When the concentration of the CdS QDs was 160 μM, the cell viability remained 85% after 24 hours, which indicate that few cadmium ions dissociated from CdS QDs [21].” Besides being poorly written and containing several mistakes, the authors cannot conclude the leaching of Cd ions into solution, subsequently reducing the cell viability. Further experiments would be needed to prove this. Several mistakes include u instead of the greek letter for micron, space bettween the reference and last word and poor English.

R4: According to the suggestion of the reviewer, “When the concentration of the CdS QDs was 160μM, the cell viability remained 85% after 24 hours, which indicate that few cadmium ions dissociated from CdS QDs [21].” Has been revised to: “ It can be seen from figure5 that the CdS QDs can  be considered to be nontoxic to HepG2 cells(concentration below 160μM ), for the viability of the HepG2 cells still exceeds 85% at the concentration of 160μM after being incubated with the CdS QDs s for 24 hour”. And the sentence: “Due to the 160 uM CdS QDs exhibit very weak toxicity for HepPG2 cells, when the concentration of CdS QDs smaller than 160 μM we can use the CdS QDs for cellular imaging.” has been revised to: “When the concentration is lower than 160 μM, the cell viability is maintained above 85%. But at 200 μM, the 24-hour cell viability was reduced to less than 70%, so we used concentrations below 160 μM in the following experiments.”.

Reviewer 2 Report

The authors have fulfilled all my requests and the paper is now suitable for publication.

Author Response

We would like to thank the comments and suggestions of the reviewer which are definitely helpful for improving the quality of our manuscript.